# Modification of Morphology and Glycan Pattern of the Oviductal Epithelium of Baboon *Papio hamadryas* during the Menstrual Cycle

**DOI:** 10.3390/ani12202769

**Published:** 2022-10-14

**Authors:** Salvatore Desantis, Maria Albrizio, Luca Lacitignola, Pietro Laricchiuta, Mario Cinone

**Affiliations:** 1Department of Emergency and Organ Transplantation (DETO), University of Bari Aldo Moro, S.P. Casamassima Km 3, 70010 Valenzano, Italy; 2Safari Zoo, 72015 Fasano, Italy

**Keywords:** fallopian tubes, glycoconjugates, histochemistry, laparoscopy, lectin, salpingectomy, sex hormones, uterine tube, vaginal cytology

## Abstract

**Simple Summary:**

The oviduct or Fallopian tube is the anatomical region where fertilization and early embryonic development take place. The morphology and activity of the epithelial cells are under the control of the sex hormone levels. The non-ciliated cells secrete oviduct-specific estrogen-dependent glycoproteins whose glycan component, despite its key role in the oviductal function, has not yet been studied in baboon species. This study examined the morphology and the glycan composition of the oviductal epithelium of the baboon *Papio hamadryas* during the menstrual cycle. Different segments of the laparoscopically removed oviducts from healthy adult females during the follicular, preovulatory, and luteal phases were processed. The stage of the menstrual cycle was based on the sex hormone levels and the vaginal cytology features. The morphological and glycopattern analyses revealed that in the epithelium lining, all the oviductal segments were taller, more differentiated, and richer in glycoconjugates during the preovulatory phase than in the follicular and luteal phases. A region-specific glycosylation pattern was also detected. These results provide an insight into the molecular differences occurring in the oviductal regions during the phases of the menstrual cycle of the baboon oviduct, which is a primate phylogenetically close to humans and whose reproductive tract is similar to that of women and influenced by the same hormonal events.

**Abstract:**

The mammalian oviduct is a highly specialized structure where fertilization and early embryonic development occur. Its mucosal epithelium is involved in maintaining and modulating a dynamic intraluminal fluid. The oviductal epithelium consists of ciliated and non-ciliated (secretory) cells whose differentiation and activity are sex hormone-dependent. In this study, we investigated for the first time both the morphology and the glycan composition of baboon oviductal epithelium during the menstrual cycle. Oviducts were laparoscopically removed from 14 healthy adult female *Papio hamadryas* whose menstrual cycle phase was assessed based on the sex hormone levels and the vaginal cytology features. Histological investigations were carried out on fimbriae, infundibulum, ampulla, and isthmus separately fixed in 4% (*v*/*v*) paraformaldehyde, embedded in paraffin wax, and stained with hematoxylin-eosin for morphological analyses and using a panel of nine fluorescent lectins for glycoconjugate characterization. The histomorphological analysis revealed that in the entire oviduct (i) the ciliated and non-ciliated cells were indistinguishable during the follicular and luteal phases, whereas they were highly differentiated during the preovulatory phase when the non-ciliated cells exhibited apical protrusions, (ii) the epithelium height was significantly higher in the preovulatory phase compared to other menstrual phases, and (iii) the number of ciliated cells significantly (*p* ≤ 0.05) increased from the fimbriae to the infundibulum and progressively reduced in the other oviductal segments with the lower presence of ciliated cells in the isthmus. The glycan characterization revealed a complex and region-specific composition during the different phases of the menstrual cycle. It can be summarized as follows: (i) high-mannosylated N-linked glycans (Con A reactivity) were present throughout the oviductal epithelium during the entire menstrual cycle and characteristically in the apical protrusions of non-ciliated cells of the ampulla during the preovulatory phase; (ii) sialoglycans with α2,3-linked sialic acids (MAL II binding) were expressed along the entire oviductal surface only during the preovulatory phase, whereas α2,6-linked ones (SNA affinity) were also detected in the surface of the luteal phase, although during the preovulatory phase they were characteristically found in the glycocalyx of the isthmus cilia, and O-linked sialoglycans with sialic acids linked to Galβl,3GalNAc (T antigen) (KsPNA) and terminal N-acetylgalactosamine (Tn antigen) (KsSBA) were found in the entire oviductal surface during all phases of the menstrual cycle; (iii) GalNAc terminating O-linked glycans (HPA staining) were mainly expressed in the entire oviducts of the luteal and preovulatory phases, and characteristically in the apical protrusions of the isthmus non-ciliated cells of the preovulatory phase; and (iv) fucosylated glycans with α1,2-linked fucose (LTA reactivity) occurred in the apical surface of fimbriae during the luteal phase, whereas α1,3/4-linked fucose (UEA I binders) were present in the apical protrusions of the ampulla non-ciliated cells and in the apical surface of isthmus during the preovulatory phase as well as in the isthmus apical surface of follicular-phase oviducts. These results demonstrate for the first time that morphological and glycan changes occur in the baboon oviductal epithelium during the menstrual cycle. Particularly, the sex hormone fluctuation affects the glycan pattern in a region-specific manner, probably related to the function of the oviductal segments. The findings add new data concerning baboons which, due to their anatomical similarity to humans, make an excellent model for female reproduction studies.

## 1. Introduction

The oviduct plays a crucial role in mammalian reproduction because it is the place in which gamete transport and maturation, fertilization, and early embryonic development occur. The oviduct consists of four regions: the fimbria, the infundibulum, the ampulla, and the isthmus. The fimbria, a fringe of finger-like structures projecting from the infundibulum, is responsible for the egg ‘pick-up’ into the oviduct after ovulation. The fimbria and infundibulum push the ovulated eggs into the ampulla, where fertilization and early embryonic development occur. The isthmus functions as a sperm reservoir and is the site where the last steps of sperm capacitation occur [1,2].

The epithelium lining the oviduct consists of ciliated and non-ciliated (secretory) cells. The epithelium of the primate oviduct is very sensitive to sex steroid hormone fluctuaction. Estradiol induces and maintains the mature epithelium of the baboon oviduct, whereas steroid withdrawal or the administration of progesterone causes regression of the epithelium. Similar to humans [3], in non-human primates, the secretory cells are maximally developed during the periovulatory phase of the menstrual cycle, and secretory activity is absent or minimal during the late luteal phase [4]. It is clearly established that estradiol induces hypertrophy, hyperplasia, and the differentiation of the mature ciliated and non-ciliated (secretory) cells, whereas progesterone, in the presence or absence of estradiol, causes atrophy, deciliation, and loss of secretory activity within the mammalian oviduct [5].

Oviductal fluid is a complex solution containing ions, energy substrates, amino acids, prostaglandins, steroid hormones as well as growth factors, and various proteins including glycoproteins, see [6,7] for references.

The secretory products of non-ciliated cells contribute to the composition of the unique intraluminal environment that plays a key role in mammalian reproduction. There are several secreted proteins in tubal fluid that contribute to embryo development and transport [8,9]. The mammalian oviduct, including the primate oviduct, synthesizes and secretes a family of estrogen-dependent, oviduct-specific secretory macromolecules including glycoproteins that appear to be present only in the apical tips of the mature secretory cells [10,11]. The oviductal glycoprotein binds to the zona pellucida and perivitelline space of ovulated oocytes and embryos within the oviduct [11]. In addition, oviductal glycoproteins (OGP) play a role in pre-fertilization reproductive events such as sperm capacitation, sperm–oocyte binding, oocyte penetration, modification of the zona pellucida, and regulation of polyspermy [12]. However, in murine species, oviductal glycoproteins could be not essential for fertility because the targeted invalidation of OGP1 in the mouse is without any consequence in female reproduction (see [13] for references).

The baboon, genus Papio, comprises several species such as *Papio ursinus* (chacma), *Papio anubis* (olive), *Papio cynocephalus* (yellow), *Papio papio* (red or Guinea), as well as *Papio hamadryas* (sacred), and the generic term “baboon” tends to be used interchangeably in many studies [14]. *Papio hamadryas* lives in subdesert, arid brushland, steppe, plain, and savannah, associated with areas containing vertical cliff faces (on which they sleep) and it is the only baboon species which has dispersed out of Africa and still inhabits Arabia [15]. The *Papio hamadryas* population is stable or increasing in the wild [16]. They form an important link in local food webs, making nutrients they obtain from plants and small animals available to larger animals. *Papio hamadryas* have a multilevel social system in which all females, regardless of reproductive state, maintain close proximity to specific leader males, resulting in the formation of one male units (OMU) [17]. It is a continuous breeder even in captivity; its menstrual cycle lasts 33.4 ± 2.1 days with menstruation lasting 3.2 ± 1.0 days. The peak estrogen serum level at the time of ovulation, 245 ± 30.5 pg/mL, is comparable to values in women. Progesterone secretion increases after the onset of LH surge and reaches a maximum level of 11.5 ± 2.3 ng/mL [18]. Due to their charm, they provide a great deal of entertainment to people who visit them in zoos. In addition, the baboon is phylogenetically close to humans [19] and represents a good translational model in medical research including reproductive biology [20].

Several reports have been published on the cyclic changes of the mucosal epithelium of some oviductal segments in non-human primates such as the cynomolgus macaques *(Macaca fascicularis*) [4] and the baboon *Papio anubis* [11,21], whereas no study has been reported on the morphological effect of the menstrual cycle on the entire oviduct of the baboon *Papio hamadryas*. In addition, the presence of oviduct-secreted specific glycoproteins has been reported in several primates such as *Papio anubis* [21], humans [22], and their estrogen-dependent secretion has been demonstrated in the baboon *Papio anubis* [10,11,21,23], and humans [24,25], but their glycan composition has not been investigated.

Glycans of oviductal epithelium play a key role in oviductal physiology and mammalian reproduction [9,26,27]. Changes in glycan patterns have been reported in the oviductal glycoproteins of several mammals due to sex hormone fluctuation as well as after experimental treatment by in vivo injection of estradiol (see [28] for references). This work contributes to the knowledge on the reproductive system of the *Papio hamadryas* by correlating the menstrual cycle stages with changes in the glycan expression of the oviduct.

## 2. Materials and Methods

### 2.1. Animals

The study was conducted on fourteen captive females of *Papio hamadryas* at the Safari Zoo (Fasano (BR), southern Italy) from November 2020 to December 2020. Salpinges used in this study were removed during a laparoscopic salpingectomy. Laparoscopy was employed as alternative to laparotomy in a clinical project for birth control in captivity authorized with a written informed consent by the Zoo’s property (Leo 3000 S.p.a, c/o Safari Zoo) and also approved by the Ethical committee of the Department of Emergency and Organs Transplantation of the University of Bari “Aldo Moro” (approval number: 05/2020). To be included in the clinical project, animals had to satisfy the following criteria: to be a (1) sexually mature female, (2) not pregnant, (3) without endometrial disorders (hyperplasia, endometritis, neoplasia). To evaluate these conditions: achievement of sexual maturity, pathologies of the reproductive tract and to exclude pregnancy, animals were anesthetized and subjected to a gynecological examination associated with uterine ultrasound. Moreover, during the laparoscopic procedure, the presence of either corpus luteum or corpus albicans was searched to discriminate the sexually mature subjects from impuberal ones. All surgical interventions were conducted in accordance with Italian law in the respect of animal welfare.

### 2.2. Evaluation of the Stage of the Menstrual Cycle

The stage of the menstrual cycle was identified on the day of surgery coupling blood sex hormone concentration and vaginal cytology evaluation. These analyses are required to properly assess the time of the menstrual cycle by means of only one blood sample in non-human primates. This procedure is adopted to also ensure maximum animal welfare avoiding repeated sedations.

### 2.3. Hormonal Assays

Blood samples were recovered from the cephalic vein. Serum samples were separated from the clot by 15 min centrifugation at 2000× *g* and stored at −80 °C until use. Beta-estradiol and progesterone were measured using monkey ELISA kits purchased from My BioSource (San Diego, CA, USA). The lower detection limit was 40 pg/mL for beta estradiol and 0.12 ng/mL for progesterone. Measurements were made in duplicate.

### 2.4. Vaginal Cytology

Vaginal cytology was used to determine the stage of the baboon menstrual cycle. Vaginal epithelial cells were gently collected from the anterior vagina of sedated animals by a moistened vaginal swab. Cells were then rolled onto microscope slides, fixed immediately and stained by Harris–Shorr method and observed under an optical microscope (Nikon Eclipse E600, Nikon, Tokyo, Japan). Images captured by a digital camera (Digital sight DS Fi2, Nikon, Tokyo, Japan). Presence and proportion of epithelial cells with nuclei (parabasal, intermediate, superficial cells), cornified cells (without nuclei), leukocytes, erythrocytes and bacteria were observed and evaluated to identify the stage of the reproductive cycle. At least 100 epithelial cells were counted, and the cornification index (CI) evaluated as the number of cornified cells ×100/total number of epithelial cells. A CI value > 80% was associated with the preovulatory phase [29,30].

### 2.5. Sampling and Tissue Preparation for Histological Analysis

Oviducts (Figure 1) from follicular (n = 4), preovulatory (n = 5), and luteal (n = 5) phases of baboons *Papio hamadryas* immediately after removal were immersed in 4% (*w*/*v*) phosphate-buffered (PBS) paraformaldehyde and transported to the laboratory at room temperature (RT).

The phase classification of the menstrual cycle was performed by hormonal assay and vaginal cytology findings. The oviducts were then trimmed of excess tissue and the infundibulum with fimbriae, ampulla, and isthmus were separated. The anatomical regions were identified because the finger-like fimbrial zone continues with the infundibulum that has a ductal appearance. The ampulla is the longest oviductal region with the largest diameter. Lastly, the isthmus is the caudal region of the oviduct located near the utero-tubal junction. The latter segment was not investigated in the present study because it was damaged during the salpingectomy. The tissues were fixed for 24 h at RT, dehydrated in an ethanol series, cleared in xylene, and embedded in paraffin wax. For each phase of the menstrual cycle, the oviductal segments of two samples were longitudinally oriented in the embedding blocks, whereas the remaining specimens were oriented crosswise. Serial sections (5 μm thick) were cut and, after de-waxing with xylene and hydration in an ethanol series of descending concentrations, were stained with hematoxylin-eosin for morphological analyses and by means of the lectin histochemistry for glycoconjugate characterization.

### 2.6. Lectin Histochemistry

Tissue sections were rinsed in 0.05 M Tris-HCl-buffered saline (TBS), pH 7.4, and incubate at RT for 1 h in the dark with appropriate dilutions of nine fluorescent lectins (Table 1) diluted in the TBS. All lectins were obtained from Vector Laboratories (Burlingame, CA, USA) except for MAL II that was purchased by Glycomatrix (Dublin, OH, USA). After three rinses in TBS, slides were mounted in Fluoroshield with DAPI (Sigma-Aldrich, St. Louis, MO, USA).

Each experiment was repeated twice for each sample. Controls for lectin staining included (1) substitution of the substrate medium with buffer without lectin and (2) incubation with each lectin in the presence of its hapten sugar (0.5 M). All control experiments gave negative results (Figure 2). The evaluation of staining intensities was based on subjective estimates of three of the authors (S.D., M.A., M.C.), and the inter- and intraobserver error was tested to assess the reproducibility of the system. A high degree of consistency was found among observers.

### 2.7. Sialidase Treatment

Demonstration of sialic acids was carried out according to [31]. Briefly, some sections were digested at 37 °C for 16 h with 0.86 U sialidase (Type V, from Clostridium perfringens; Sigma-Aldrich) in 0.1 M sodium acetate buffer pH 5.5 containing 10 mM CaCl2 and thereafter subjected to the staining procedures described above with the lectins MAL II, SNA, PNA, and SBA. Prior to the neuraminidase treatment, a saponification technique was performed to render the enzyme digestion effective, with 0.5% KOH in 70% ethanol for 15 min at RT. As controls of the enzymatic digestion, some sections were incubated in enzyme-free buffer under the same duration and temperature conditions. In control sections, cleavage of sialic acid was not evident.

### 2.8. Morphometrical Analysis

The height of epithelium lining the oviducts and the number of ciliated cells were determined using 15 microphotograph fields randomly detected and taken with a digital camera (DS-U3, Nikon, Japan) connected to a light microscope Eclipse Ni-U (Nikon, Japan), using a 100× objective. The images were analyzed by the image-analyzing program NIS Elements BR (Vers. 4.20) (Nikon, Japan). For the measurement, cells in which the plane of the section clearly passed through the cell nucleus, parallel to the longitudinal axis of the cells were selected.

### 2.9. Statistical Analysis

To evaluate the statistical significance of data, the Statistical Package for Social Science (SPSS, version 19) software was employed. The Kolmogorov–Smirnov test was used to evaluate the normal distribution of data. Comparisons among different periods of the menstrual cycle were performed by the one-way ANOVA test and Bonferroni post hoc test. *p* values were two tailed and a *p* value < 0.05 was considered significant. All values were expressed as mean ± standard deviation (S.D.).

## 3. Results

### 3.1. Steroid Hormones

Measurements of circulating levels of beta-estradiol and progesterone evidenced that nine out of fourteen subjects were in the follicular phase of their menstrual cycle, and the remaining five baboons were in the luteal phase. The mean value of beta-estradiol blood concentration significantly (*p* < 0.01) increased from the follicular phase to the preovulatory phase (96.54 ± 18.7 pg/mL vs. 193.55 ± 71.8 pg/mL) and significantly *p* < 0.001) decreased in the luteal phase (193.55 ± 71.8 pg/mL vs. 50.11 ± 19.9 pg/mL) when reached the lowest level of the menstrual cycle (Figure 3). Progesterone concentrations were 0.239 ± 0.1 ng/mL in the follicular phase, 0.355 ± 0.07 ng/mL in the preovulatory phase and reached the highest levels (4.0 ± 0.5 ng/mL) in the luteal phase (follicular phase vs. luteal phase *p* < 0.001; preovulatory phase vs. luteal phase *p* < 0.001 (Figure 3).

### 3.2. Vaginal Cytology

The evaluation of the morphological features of the squamous stratified vaginal epithelium of the animals scheduled in follicular, preovulatory, or luteal phase from the steroid hormone assay displayed phase-specific features. In the follicular phase, the vaginal smears showed predominant large intermediate cells with a partial basophilic cytoplasm (Figure 4A); in the preovulatory phase, the smears displayed superficial, keratinized cells with or without a pyknotic nucleus with a CI index > 80% (Figure 4B); lastly, vaginal smears from luteal-phase samples were characterized by the presence of small nucleated parabasal cells (Figure 4C).

### 3.3. Histological Analysis

The histological sections (Figure 5) showed that except for the fimbriae, which are finger-like mucosal folds, the oviduct was constituted of mucosal folds, a muscular wall and connective serosa. The mucosal folds were more numerous and branched in the ampulla compared to other regions of the oviduct. In addition, the ampulla is the region with the largest diameter of the lumen. The muscular wall was thin in the ampulla, whereas it was thick in the infundibulum and the isthmus.

The epithelium lining the mucosal folds is constituted of ciliated cells and non-ciliated (secretory) cells. Morphological analysis revealed that the epithelium in the follicular and luteal phases consisted of a single layer of cuboidal cells essentially devoid of cilia thus the ciliated and non-ciliated cells were indistinguishable (Figure 6A,C). On the contrary, the ciliated and non-ciliated cells were clearly distinguishable during the preovulatory phase when the highly differentiated epithelium was columnar in shape and the non-ciliated cells exhibited apical protrusions (Figure 6B).

Morphometric analysis revealed a clear change in the epithelium height of all oviductal segments during the menstrual cycle with an increase from the follicular phase to the preovulatory phase and a decrease in the luteal phase (Figure 7).

A comparison of the epithelium height of each oviductal segment during the menstrual cycle (Figure 8) evidenced the following: (1) the fimbriae epithelium significantly (*p* < 0.001) increases from follicular to preovulatory phase (11.45 ± 1.77 µm vs. 23.37 ± 2.36 µm) and decreases (*p* < 0.001) in the luteal phase (11.85 ± 1.68 µm), when similar values of the follicular phase ones were reached; (2) the infundibulum epithelium significantly (*p* < 0.001) increases from the follicular phase to preovulatory phase (10.69 ± 2.28 µm vs. 27.97 ± 3.24 µm) and decreases (*p* < 0.001) in the luteal phase (15.15 ± 2.24 µm), remaining significantly (*p* < 0.001) higher than follicular phase (10.69 ± 2.28 µm vs. 15.15 ± 2.24 µm); (3) the same trend displays the ampulla epithelium as being 13.94 ± 2.36 µm, 30.05 ± 4.02 µm, and 20.89 ± 3.57 µm in the follicular, preovulatory, and luteal phases, respectively; and (4) the isthmus epithelium significantly (*p* < 0.001) increases from the follicular phase to the preovulatory phase (19.87 ± 4.04 µm vs. 28.77 ± 3.39 µm), and decreases (*p* < 0.001) in the luteal phase (16.11 ± 1.58 µm), reaching the lower value (*p* < 0.01) of the menstrual cycle at 19.87 ± 4.04 µm in the follicular phase and 16.11 ± 1.58 µm in the luteal phase.

The percentage of the ciliated cells significantly increased from fimbriae to infundibulum (0.85 ± 3.67% vs. 54.96 ± 2.49%) and progressively decreased, although with no statistical significance, in the ampulla (53.305 ± 6.10%) and isthmus (49.52 ± 7.9%) (Figure 9).

### 3.4. Lectin Histochemistry

The results of the lectin staining pattern of the mucosal epithelium of the baboon *Papio hamadryas* are summarized in Table 2.

### 3.5. High-Mannose Glycans

Con A, specific to high-mannose N-linked glycans, stained the cytoplasm, including the apical cytoplasm of the entire oviductal epithelium as well as the basal lamina during the follicular (Figure 10A) and luteal (Figure 10C) phases. The oviductal epithelium from preovulatory-phase animals displayed apical reactivity with Con A, which linked apical protrusions of non-ciliated cells in the ampulla segment (Figure 10B). The basal lamina gave a negative reaction throughout the entire oviductal epithelium during the preovulatory phase.

### 3.6. Sialoglycans

MAL II, specific to α2,3-sialoglycans, did not react with the epithelium lining the entire oviduct during the follicular and luteal phases, whereas it bound the apical surface of the entire oviduct during the preovulatory phase (Figure 11A).

### 3.7. Detection of the Terminal and Sialic Acid Penultimate T-Antigen with PNA and KOH-Sialidase (s)PNA

PNA, specific to terminal Galβl,3GalNAc dimer (T-antigen), did not stain the epithelium lining the mucosal oviduct of all investigated baboons (Figure 12A). Cleavage of sialic acids with KOH-sialidase pretreatment revealed cryptic PNA-binding sites in the apical surface of all oviductal segments during the entire menstrual cycle (Figure 12B–F). During the follicular phase the apical surface showed a thinner positivity (Figure 12B,C) when compared with the other phase of the menstrual cycle (Figure 12D–F).

### 3.8. Fucosylated Glycans

LTA, specific to terminal αL-Fucose, weakly bound the cytoplasm and the apical surface of the isthmus during the follicular phase (Figure 13A). In the preovulatory-phase baboons, LTA reacted with the apical blebs of the non-ciliated cells in the ampulla (Figure 13B) and with the apical surface in the isthmus. No LTA binding sites were found in the oviductal epithelium of luteal-phase animals.

UEA I, specific to α1,2-linked fucose, only reacted with the apical surface of fimbriae during the luteal phase (Figure 13C).

### 3.9. GalNAc-Terminating Glycans

HPA, specific to terminal GalNAcα-Ser/Thr (Tn antigen), gave a weak reaction with the epithelial luminal surface and cytoplasm of the fimbriae during the follicular phase (Figure 14A). This lectin showed much more binding sites during the preovulatory phase. Fluorescence signals were observed as a granular pattern in the apical zone of the epithelium lining the entire oviduct, with a lower binding site in the fimbriae (Figure 14B) than in the other oviductal zones (Figure 14C). During the preovulatory phase, the isthmus displayed HPA positivity of the apical blebs (Figure 14D). During the luteal phase, HPA binding sites were found on the luminal surface of the entire oviduct. The fimbriae and isthmus (Figure 14E) zones displayed fewer binding sites than the infundibulum and ampulla (Figure 14F).

DBA binding sites, showing terminal αGalNAc, were not observed in the oviductal epithelium of the follicular-phase baboons. This lectin showed a very faint reactivity with the luminal surface of the fimbriae during the preovulatory phase (Figure 14G) and with the luminal surface of the infundibulum and the isthmus during the luteal phase (Figure 14H).

SBA, specific to terminal α/βGalNAc, did not bind the oviductal epithelium of the investigated samples (Figure 14I). Cleavage of the sialic acid with KOH-sialidase pretreatment produced a weak staining of the follicular-phase oviduct (Figure 14L) and a stronger reactivity in the preovulatory oviduct (Figure 14M,N). The SBA affinity of oviducts from luteal-phase baboons was not affected by KOH-sialidase treatment.

## 4. Discussion

This study describes, for the first time, the morphological and glycan changes occurring in the oviductal epithelium of *Papio hamadryas* during the menstrual cycle. The results were supported by the changes in sex hormones and vaginal cytology occurring during the menstrual cycle.

The fluctuations of sex hormones determine the primate menstrual cycle and regulate the morphology and the activity of the female reproductive apparatus (see [5] for references). The baboon female reaches puberty at the age of 4.3 years. The average duration of the menstrual cycle is 32 days (from 24 to 38 days) and the gestation lasts 172 days [32]. Each phase of the menstrual cycle is characterized by a specific blood concentration of sex hormones. Beta-estradiol is the protagonist of the preovulatory phase, on the contrary progesterone plays a major role in the luteal phase. During the menstrual cycle, the sex hormonal pattern is similar in *Papio hamadryas* and woman even if the serum concentration is lower in the baboon [33]. The plasma levels of estradiol and progesterone measured in the present study matched with the radioimmunoassay results reported in baboons *Papio hamadryas* during the phases of the menstrual cycle [33]. The cytological feature of the vaginal smear is also linked to the phases of the menstrual cycle [34]. In response to the level of estrogen in the blood, the endometrium proliferates and the vaginal epithelial cells cornificate so that changes in cell typology are indicative of underlying endocrine events and can characterize the stages of the menstrual cycle in this species [34] as well as in woman [35]. Parabasal basophil cells, a feature of the luteal phase, pass throughout large intermediate cells until becoming superficial cornified acidophil cells in the preovulatory phase.

The morphometric analysis revealed that the *Papio hamadryas* epithelium lining all the oviductal segments during the preovulatory phase were taller than in the other menstrual phases. The comparison of the luteal and follicular phases revealed that the epithelium of the infundibulum and the ampulla was taller in the luteal phase, and shorter in the isthmus of the luteal phase, whereas it did not differ in the fimbriae. This finding suggests a marked regional difference in the response of the lining epithelium to the serum concentrations of sex hormones. Morphometric analysis of the epithelium lining the entire oviduct in other species of Papio has not been carried out. Verhage et al. [21] measured the cell height of the baboon *Papio anubis* ampulla and isthmus but not of fimbriae and infundibulum during the menstrual cycle. The comparison between *Papio hamadryas* and *Papio anubis* morphometric data reveals that the epithelium height of the ampulla and isthmus has the same cyclic trend because the cell height increases from the follicular to the preovulatory phase and decreases in the luteal phase. In *Papio anubis* oviduct, the shortest epithelium was detected in the luteal phase when the ampulla and isthmus epithelium decreased by about 50% and 18%, respectively, when compared with the preovulatory phase [21]. However, the *Papio hamadryas* showed a shorter epithelium in the ampulla during the follicular phase (31 µm vs. 14 µm) and the preovulatory phase (35 µm vs. 30 µm), as well as in the isthmus during the entire menstrual cycle (36 µm vs. 19 µm in the follicular phase, 39 µm vs. 28 µm in the preovulatory phase, and 32 vs. 16 µm in the luteal phase) when compared with *Papio anubis*. The observed differences confirm that the height of the oviductal epithelium varies among the species as it has been observed in other mammals [36,37], including primates [4].

Morphological analysis revealed that the epithelium of the baboon *Papio hamadryas* oviduct showed changes from the fimbriae to isthmus segments during the menstrual cycle. The epithelium displayed a regressed state, characterized by low height and a single layer of cuboidal cells with no evidence of cilia and secretory activity during the follicular and luteal phase. On the contrary, the epithelium showed a morphological mature state during the preovulatory phase when the oviductal mucosa was lined by a single-layered columnar epithelium with fully differentiated ciliated and non-ciliated cells. The non-ciliated cells were characterized by apical protrusions. In this phase, the epithelium produces substances that provide the optimum environment for the free-floating unfertilized oocyte and that also facilitate the fertilizing capability of spermatozoa [11]. The observed morphological features are related to the sex hormone concentration. Specifically, beta-estradiol blood levels significantly (*p* < 0.01) increased from the follicular phase to the preovulatory phase and showed the lowest levels (*p* < 0.001) in the luteal phase, whereas the progesterone levels peaked in this latter phase (*p* < 0.001). The steroid control of the oviductal epithelium of the baboons has been demonstrated in ovariectomized and sex hormone-treated *Papio anubis* [11,21,24]. The epithelium lining the ampulla and isthmus in the ovariectomized baboon consisted of a single cuboidal cell type containing neither secretory granules nor cilia. The epithelium from both normal late follicular animals and ovariectomized animals treated with estradiol for 14 days displayed a mature condition characterized by fully differentiated non-ciliated (secretory) cells and ciliated cells. Oviducts from animals treated with estradiol for 14 days and then treated with estradiol and progesterone for another 14 days displayed again cuboidal undifferentiated cells, neither secretory nor ciliated cells. Similar morphological changes during the menstrual cycle have been reported in the oviductal epithelium of the non-human primate Cynomolgus macaques (*Macaca fascicularis*) [4].

The count of ciliated cells was carried out during the preovulatory phase when they were clearly distinguishable. The number of ciliated cells significantly (*p* ≤ 0.05) increased from the fimbriae to the infundibulum and progressively reduced in the other oviductal segments with the lowest presence of ciliated cells in the isthmus. This trend has also been reported in other mammals such as estrus golden hamsters and follicular-phase bovine [24], and estrus mare [38]. As concerns the baboons, a study on the ampulla and isthmus of *Papio anubis* [21] confirms our findings. Unfortunately, no data are available on the number of ciliated cells in the baboon fimbriae and infundibulum. However, considering that the cumulus-oocyte complexes (COCs) are ovulated into the peritoneal cavity of primates, the ciliated cells of the fimbriae create a current of peritoneal fluid toward the oviductal ostium, facilitating oocyte passage into the oviductal tube [39]. The increase in ciliated cells in the infundibulum, and consequently the increased cilia beating in this segment, could be responsible for creating a current flow that helps the COC movement towards the ampulla. Recently it has been demonstrated that functional motile cilia in the infundibulum are essential for picking up the ovulated oocytes [40].

Lectin histochemistry demonstrated region-specific glycan pattern during the phases of the menstrual cycle. The fimbriae reacted with Con A, HPA, and sPNA during the entire menstrual cycle showing the constant presence of both high-mannose N-linked glycans [41] and O-linked glycans terminating with Tn antigen (GalNAcα-Ser/Thr) [42] and sialylated Galβl,3GalNAc (T antigen) [41]. As for N-glycans, despite the extensive presence in this and the other tracts of the oviductal epithelium, their function is not known, although N-glycans could contribute to the biological properties of the glycoproteins in terms of structure, activity, susceptibility to protease, and antigenicity [43]. The diffuse presence of the Tn antigen could represent an intermediate step in the synthesis of the O-linked (mucin-type) glycans [42]. The presence of sialoglycans on the apical surface with subterminal GalNAc (sSBA) revealed an increasing amount from the follicular to the preovulatory phase in the fimbriae as well as in all other segments of the oviduct, whereas they were absent during the luteal phase. Sialic acids have diverse functional roles such as repulsion of cell–cell interaction, protein stabilization ion binding, and ion transport [44]. Therefore, this finding suggests that the sialoglycans are linked to the morpho-functional state of the oviduct during the menstrual cycle. During the preovulatory phase, the presence of a more complex glycan pattern was observed because the apical surface displayed additional GalNAc-terminating O-linked glycans (DBA binding) and sialoglycans terminating with NeuNAcα2,3Galβ1,3GalNAc and Neu5Acα2,6Gal/GalNAc revealed with MAL II and SNA, respectively [45], as well as sialoglycans detected with sPNA and sSBA. The observed granular reactivity with HPA in the apical zone of this segment as well as in the infundibulum and the ampulla during the preovulatory phase could be interpreted as secretory granules. This suggests that the secretion of O-linked mucin-type glycans occurs in the baboon *Papio hamadryas* oviduct. A similar staining pattern has been observed in the human fallopian tube by means of PNA, suggesting the secretion of O-linked glycans terminating with the Galβl,3GalNAc dimer [46]. Since in the present study the oviductal epithelium of the baboon *Papio hamadryas* did not react with PNA, we can suppose that the molecular content of the oviductal secretory granules differs between these species. The glycopattern changed in the luteal phase when α2,3-linked sialic acids disappeared and αl,2fucosylated glycans (UEA I reactivity) [41] were detected. A different glycan pattern has been observed in the oviductal fimbriae of the monkey *Cebus apella* during the luteal phase because of SNA, MAA, UEA I, DBA, and SBA binding sites were detected [47]. Although the above-mentioned study did not investigate the lectin binding pattern of the monkey *Cebus apella* in the other phases of the menstrual cycle, it is possible to infer a species-specific glycan composition of the oviductal fimbriae in primates. Synthesis and production of OGP are under the sex-hormone control. In vitro experiments demonstrated that in the baboon *Papio anubis* an acidic glycoprotein is dominantly secreted in the oviductal fimbriae at the midcycle of the intact and in estradiol-treated animals [24]. In addition, the measurement of the oviductal glycoprotein mRNA revealed that the fimbriae of *Papio anubis* expresses elevated levels of the mRNA after estradiol treatment and in the late follicular stage of the menstrual cycle, whereas the oviductal glycoprotein mRNA was not detectable in the late luteal stage or in progesterone-treated baboons [23]. The presence of OGP has also been detected in the fimbriae of mouse oviducts [48]. This OGP is presumably secreted and associated with the newly ovulated oocyte [48]. Thus, fimbriae glycoproteins could have a role in the early cross-talk between the ovulated oocytes and the oviduct as well as in the transport of eggs to the infundibulum. The oviduct-specific glycoprotein has also been localized in the fimbrial epithelium at the follicular phase of the estrous cycle in bovines [37].

The infundibulum epithelium displayed the presence of high-mannose N-linked glycans (Con A binding sites) in the apical cytoplasm and sialic acid penultimate Galβl,3GalNAc and GalNAc (sPNA, sSBA) in the apical surface during the follicular phase. As in the fimbriae, the infundibulum epithelium showed a major complexity in the glycan pattern during the other phases of the menstrual cycle. In the preovulatory phase, Tn antigen (HPA) and α2,3-linked sialic acids (MAL II) appeared in the apical cytoplasm and the apical surface, respectively. The latter enriched in sialic acids as revealed by the increased sPNA and sSBA staining intensity. In addition, the presence of sialoglycans terminating with α2,6-linked sialic acids (SNA) were detected from supranuclear to the apical surface of non-ciliated cells. Since glycan synthesis apparatus occurs in the supranuclear region of the oviductal epithelial cells [21], the observed lectin reactivities in the apical cytoplasm could indicate the presence of fully differentiated merocrine secretory non-ciliated cells during the preovulatory phase. Oviductal glycoproteins have been detected within putative secretory granules of the oviductal non-ciliated cells in several mammals, including the baboon *Papio anubis* [21]. The glycan pattern of the infundibulum epithelium changed during the luteal phase when a few glycans terminating with GalNac (DBA binding) appeared on the apical surface which lost the sialic acids revealed with MAL II and sSBA. Compared to the fimbriae, the infundibulum expressed only Con A binding sites during the follicular phase and did not contain DBA binding sites during the preovulatory phase, which, instead, were expressed during the luteal phase. This finding suggests that the infundibulum could be responsible for a new luminal environment in which the ovulated oocytes find themselves during the flow to the ampulla. This binding patterns may reflect the end maturation of the oocyte in the infundibulum. Fucosylated glycans were never expressed in the infundibulum during the menstrual cycle. This finding indicates the absence of fucosyltransferase in the infundibulum of the baboon *Papio hamadryas*. It is impossible to compare our results with others as the cycle effect on the oviductal infundibulum in other primates has not been investigated. The absence of fucose residues has also been observed in the oviductal infundibulum of porcine [49] and plains viscacha (*Lagostomus maximus*) [50]. Distinct OGP has been detected in the infundibulum of several mammals [26]. More recently, it has been observed that the glycans constituting the infundibulum OGPs associate with sperm, zona pellucida and perivitelline space of mouse oocytes [48].

The ampulla epithelium displayed supranuclear high-mannose N-linked glycans (Con A reactivity) during the entire menstrual cycle. Interestingly, these glycans were also present in the apical blebs during the preovulatory phase. This is suggestive of a high production of N-linked glycans. The apical protrusions also contained LTA binding sites, suggesting the presence of secretory fucosylated glycans. The observation of differing staining with LTA and UEA I, both specific to α-L-fucose-containing glycans, depends on the specific saccharide configuration of α-L-fucose in the fucosylated glycans [35]. In the present study, we demonstrate that the baboon oviduct could secrete fucosylated N-linked glycoproteins. Additionally, during the menstrual cycle, we observed the presence of α2,6-linked sialic acids (SNA) on the apical surface. As in the above analyzed oviductal segments, the ampulla of the preovulatory phase also expressed supranuclear Tn antigen (HPA) and an increase in cell surface sialic acids (MAL II, sPNA, sSBA). A high presence of sialoglycans has been revealed in the sheep oviductal ampulla [28] and in growth phase of the camel oviduct [36]. Jaffe et al. [23] found the oviductal glycoprotein mRNA levels were elevated in the ampulla after estradiol treatment and in the late follicular stage of the menstrual cycle. Since α2,3-linked sialic acids and fucose residues were absent during the luteal phase, these carbohydrate residues could be typical moieties of the preovulatory-phase oviduct. The comparison with lectin histochemical findings in other primates suggests that the observed glycan pattern is typical of the investigated primate species. The human oviductal ampulla expresses Con A, HPA, LTA, UEA binding sites during the menstrual cycle [51], and the monkey *Cebus apella* oviductal ampulla contains N-linked glycans, MAL II, SNA, DBA, SBA, and UEA I binding sites during the luteal phase [47].

The isthmus epithelium displayed the same high-mannose N-linked glycans (Con A) distribution observed in the other oviductal segments during the menstrual cycle. The surface of this oviductal zone also expressed sialic acids (SNA, sPNA, sBA) and fusosylated glycans (LTA) during the follicular phase. During the preovulatory phase the isthmus epithelium displayed Tn antigen (HPA) in the apical blebs of non-ciliated cells, the presence of fucosylated glycans (LTA) and the increase in sialoglycans including α2,3-linked sialic acids (MAL II) and sialoderivatives revealed with sPNA and sSBA in the apical surface, and α2,6-linked sialic acids (SNA) covering the cilia. Several studies demonstrated that the surface glycocalyx plays a key role in the function of the isthmus epithelium as a sperm reservoir in mammals. Fucosylated glycans of the apical surface of the bovine isthmus oviduct are involved in a specific interaction between bovine sperm and oviduct epithelium [52] and in retaining sperm for reservoir formation and for extending sperm lifespan [27]. Moreover, porcine sperm bind to specific sialylated N-linked glycans to form the oviduct reservoir [53] and to regulate sperm Ca^2+^ influx, capacitation, and sperm lifespan [54].

As for SNA positivity of cilia glycocalyx, α2,6-sialoglycans have been detected in the isthmus of camel [36], porcine [53], horse [31], and sheep [55] oviduct as well as in human airway epithelial cells [56]. The presence of sialic acid in the cilia glycocalyx is typically observed in the isthmus of several other mammals [see 55 for references], including primates such as humans [46] and monkey [47]. Sialoglycoconjugates provide negative charge for the ciliated cell glycocalyx. The presence of a negative charge in the glycocalyx of ciliated cells in the oviduct of mammals has been considered the basis of the electrostatic interaction between cilia and oocyte cumulus cell complexes as well as in oocyte pickup and transport [51,57]. In addition, sialic acid residues could keep the cilia separated from one other, maintaining ciliary motility [58]. Evidence of the hormone-dependent presence of sialoglycoconjugates on the luminal surface of the oviductal non-ciliated cells has been found in dromedaries [36], equines [31], and rabbits [59]. In an experimental study it has been demonstrated the estrogen-dependent in vitro synthesis of baboon (*Papio anubis*) OGP, including acidic ones [10].

The apical protrusions of the secretory cells of the isthmus contain O-glycans (mucin type) as revealed with HPA. Apical blebs of the non-ciliated cells containing estrogen-dependent oviduct-specific glycoproteins have been demonstrated in the oviductal epithelium of the baboon *Papio anubis* [11]. More recently it has been demonstrated that estradiol increases IgG galactosylation through direct activation of the B4GALT1 galactosyltransferase [60]. There is no report about the presence of lectin binding sites in the apical protrusions in the isthmus as well as in the other tracts of oviduct in primates [46,47,51]. Although the role of these secretory glycans must be investigated, it has been demonstrated that isthmic glycoproteins increase the sperm binding and zona penetration [61], sperm capacitation [62], the fertilization rates and reduce polyspermy [63,64,65]. The presence of OGP has been observed in the zona pellucida and perivitelline space of the baboon oviductal oocyte [11,61]. Moreover, OGP may interact with the embryo leading to changes in cleavage rates [22,63,65]. In addition, the lectin positivity in the apical cytoplasm and the luminal surface may be related to the secretion of glycoconjugates via the extracellular vesicles (EVs). Although these small membranous vesicles have not been described in the ultrastructural study of the baboon oviduct [21], small vesicles are distinguishable in Figure 11 that shows the apical region of a mature secretory cell in the isthmus. Extracellular vesicles play an important role in cell-to-cell communication by transferring their molecular load from one cell to another. The EV surface is coated with a cell-line dependent repertoire of glycoproteins which are required to detect cell surface receptors for EV internalization by target cells (see [66] for references). In the reproductive field, EVs secreted by the oviduct and embryos, are key players in the crucial two-way dialogue between the oviduct and the embryo [67]. In porcine, the addition of oviductal EVs to the IVF medium regulates polyspermy [68], while oviductal EVs isolated from oviductal fluid improve bovine embryo quality during in vitro culture, in terms of blastocyst rates, cell number, hatching rates, embryo cryosurvival and gene expression [69]. It has been demonstrated that the molecular composition of oviductal EVs is regulated by steroid hormones [70] and that the OVGP1-protein is contained in the oviductal EVs (see [71] for references).

The luminal surface glycocalyx of oviductal non-ciliated cells could also be involved in absorptive phenomena. The absorptive activity of the intraluminal fluids occurring in the luminal surface of mammalian oviduct is regulated by aquaporins (AQPs) [72,73]. Some isoforms of AQPs are glycosylated. AQP2, 3, 8, 9 and 10 are N-glycosylated and AQP7, 10 and 12 are O-glycosylated (reviewed by [74]). Aquaporins take part not only in water transport but also in absorption or secretion of luminal fluid metabolites, thus maintaining a suitable environment for the reproductive functions. Steroid hormones can regulate the expression of some AQPs and thus control water transport into the oviduct lumen [72,73,75]. It has been suggested that the expression of AQPs in the porcine oviduct may provide the physiological medium that sustains and enhances fertilization and early cleavage-stage embryonic development [75].

As in the ampulla, the isthmus obtained from the luteal phase lacked NeuNAcα2,3Galβ1,3GalNAc and sialyl-Tn antigen (MAL II, sSBA) as well as fucose detectable with LTA. However, on the contrary of the ampulla, the isthmus displayed few oligosaccharides terminating with GalNAc (DBA affinity) on the apical surface. The observed lectin binding pattern reveals that *Papio hamadryas* isthmus expresses a species- specific glycan composition since the α2,3-linked sialic acids (MAL II), αl,2-linked fucose (UEA I), and SBA binding sites are absent when compared with the monkey *Cebus apella* oviductal isthmus [47].

## 5. Conclusions

In conclusion, this study reports, for the first time, the regional changes in the morphology and the glycosylation pattern of the lining epithelium in the baboon *Papio hamadryas* oviduct during menstrual cycle. The phases of the menstrual cycle were characterized by both the blood concentration of sex hormones and the cytological feature of the vaginal smear. The glycan pattern differences could be related to the region-specific functions. This study provides an insight into molecular differences of the baboon oviductal regions. Although this study does not provide information concerning the functional role of detected glycans, it adds further data on the glycoproteins expressed in the oviduct of a primate phylogenetically close to humans that is an excellent model for reproductive research as the reproductive tract is similar to that of women and influenced by the same hormonal events [19].

## Figures and Tables

**Figure 1 animals-12-02769-f001:**
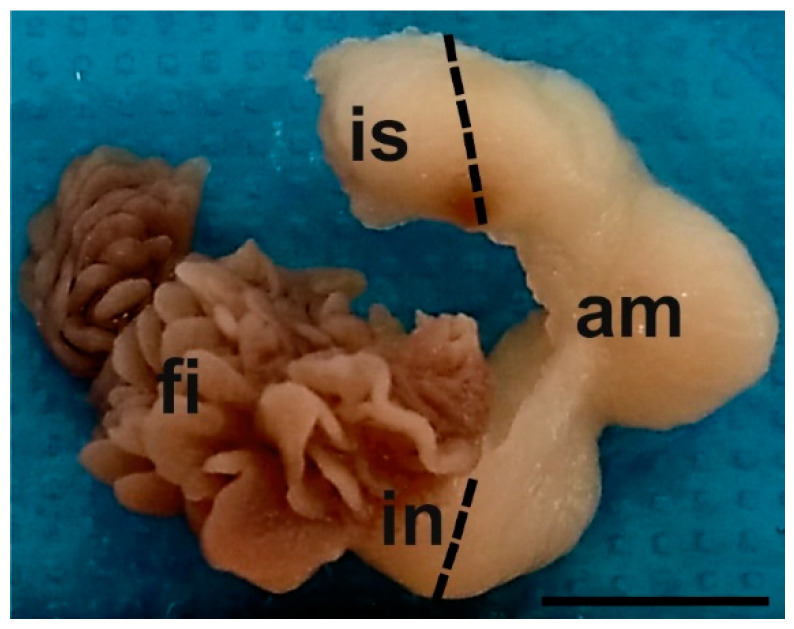
Gross anatomy of the baboon *Papio hamadryas* oviduct. Key: am, ampulla; fi, fimbriae; in, infundibulum; is, isthmus. Scale bar = 0.5 cm.

**Figure 2 animals-12-02769-f002:**
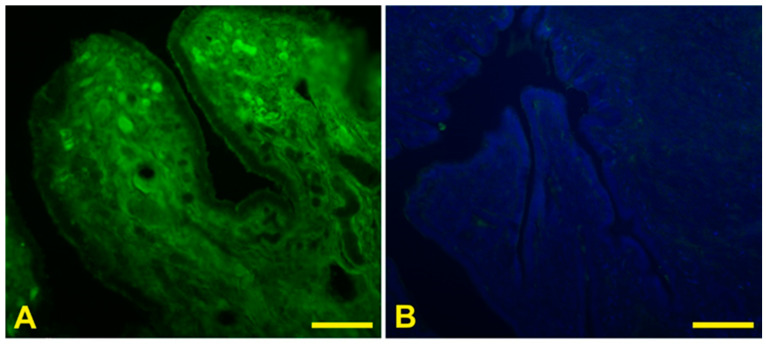
Representative effect of the hapten sugar on the lectin staining. Ampulla from preovulatory baboon *Papio hamadryas* incubated with SNA (**A**) and with SNA in presence of its hapten sugar (**B**). Scale bars: (**A**,**B**) = 100 µm.

**Figure 3 animals-12-02769-f003:**
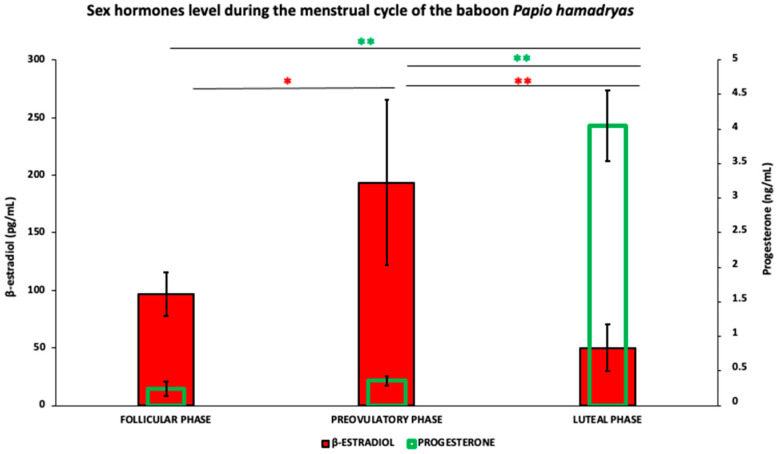
Serum concentration of beta-estradiol and progesterone in the baboon *Papio hamadryas* during the menstrual cycle. Values were expressed as means ± standard deviation (S.D.). * = *p* < 0.01; ** = *p* < 0.001.

**Figure 4 animals-12-02769-f004:**
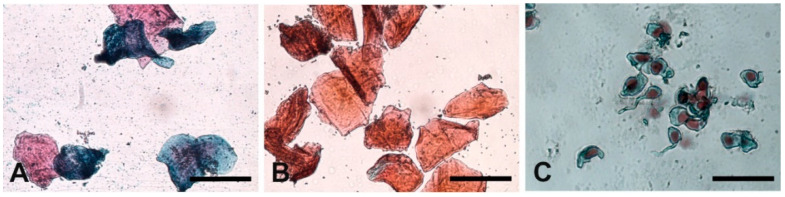
Vaginal cytology feature of the baboon *Papio hamadryas* during the menstrual cycle. (**A**) follicular phase; (**B**) preovulatory phase; (**C**) luteal phase. Harris–Shorr staining. Scale bars: (**A**–**C**) = 50 µm.

**Figure 5 animals-12-02769-f005:**
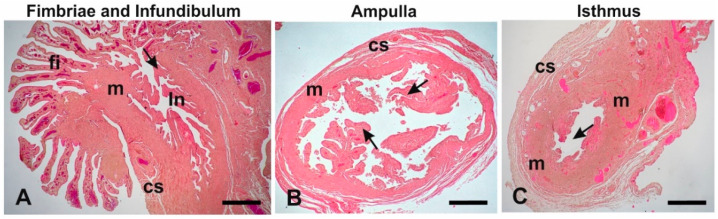
Light micrographs showing the baboon *Papio hamadryas* oviduct regions at low magnification. (**A**) Fimbriae and infundibulum; (**B**) ampulla; (**C**) isthmus. Key: cs, connective serosa; fi, fimbriae; In, infundibulum; m, muscular wall; arrow, mucosal folds. Hematoxylin-eosin staining. Scale bars: (**A**–**C**) = 500 µm.

**Figure 6 animals-12-02769-f006:**
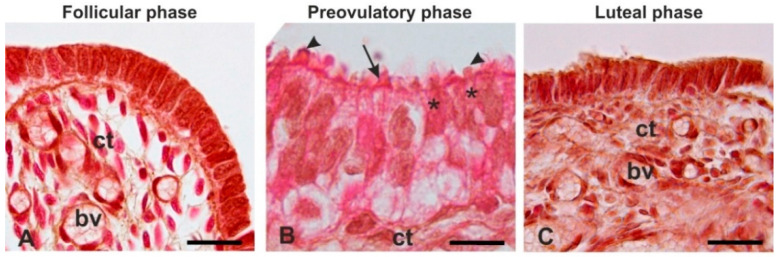
Representative light micrographs showing the morphological changes of the baboon *Papio hamadryas* oviductal epithelium during the follicular (**A**), pre-ovulatory (**B**), and luteal (**C**) phases. Note that the ciliated and non-ciliated cells were undistinguishable during the follicular and the luteal phase. Key: bv, blood vessel; ct, connective tissue; arrow, cilia; arrowhead, apical bleb; asterisk, non-ciliated cell. Hematoxylin-eosin staining. Scale bars = 10 µm.

**Figure 7 animals-12-02769-f007:**
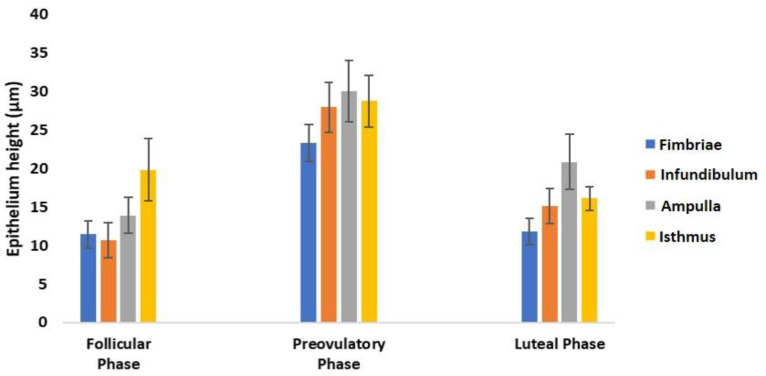
Height of the oviductal epithelium of the baboon *Papio hamadryas* during the menstrual cycle. Values were expressed as means ± standard deviation (S.D.).

**Figure 8 animals-12-02769-f008:**
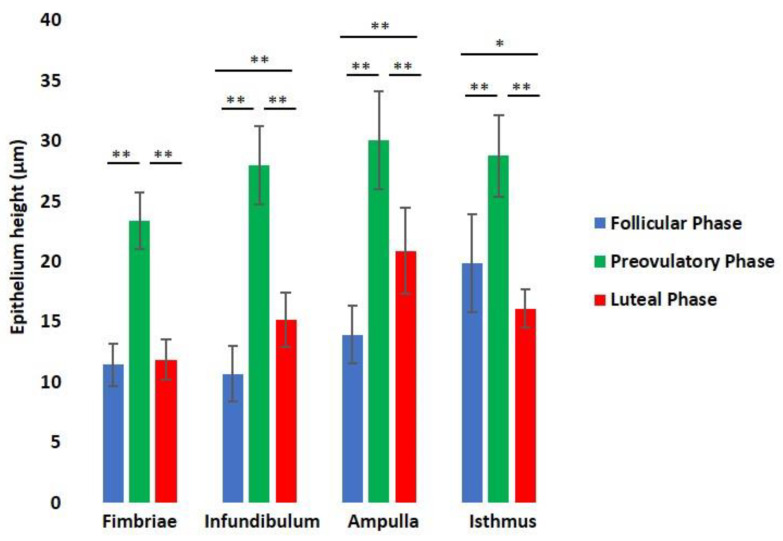
Comparison of the epithelium height of fimbriae, infundibulum ampulla, and isthmus during the menstrual cycle of the baboon *Papio hamadryas*. Values were expressed as means ± standard deviation (S.D.). Statistical differences: *, *p* ≤ 0.01; **, *p* ≤ 0.001.

**Figure 9 animals-12-02769-f009:**
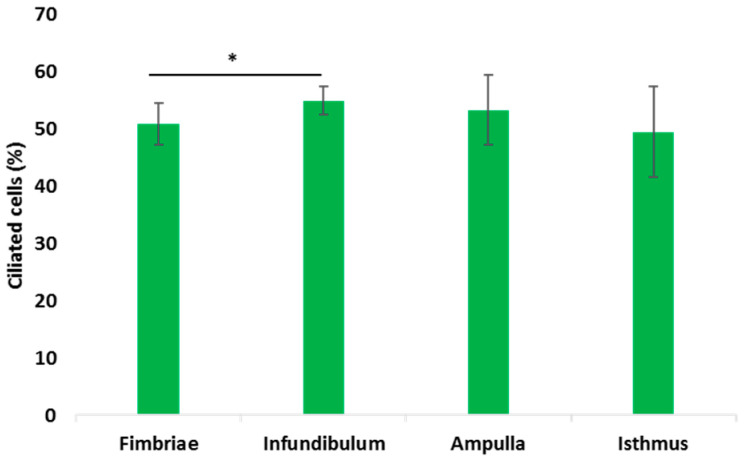
The mean percentages of ciliated cells in the epithelium of the baboon *Papio hamadryas* during the preovulatory phase. Values were expressed as means ± standard deviation (S.D.). Statistical differences: *, *p* ≤ 0.05.

**Figure 10 animals-12-02769-f010:**
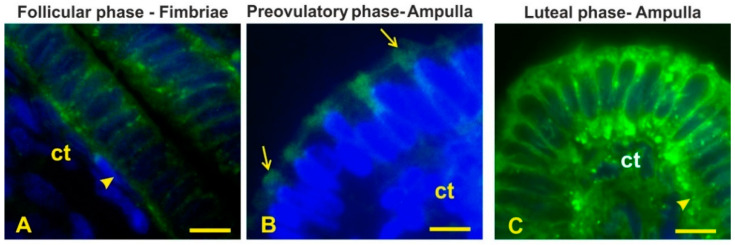
Mannosylated glycan localization as identified by Con A staining in the baboon *Papio hamadryas* oviductal epithelium during the follicular (**A**), preovulatory (**B**), and luteal (**C**) phases. (**A**) Con A staining of fimbriae/infundibulum during the follicular phase. (**B**) Con A binding sites in apical protrusion of non-ciliated cells of the ampulla during the preovulatory phase. (**C**) Con A reactivity of ampulla during the luteal phase. Key: ct, connective tissue; arrow, apical protrusion; arrowhead, basal lamina. Scale bars: (**A**) = 10 µm; (**B**,**C**) = 13 µm.

**Figure 11 animals-12-02769-f011:**
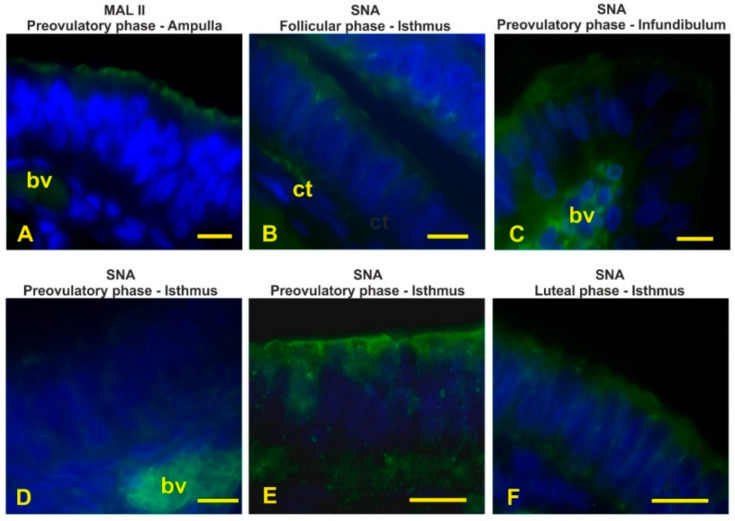
Direct detection of sialic acids with MAL II (**A**) and SNA (**B**–**F**) in the baboon *Papio hamadryas* oviductal epithelium. (**A**) MAL II staining of the ampulla during the preovulatory phase. (**B**) SNA staining of the isthmus during the follicular phase. (**C**) SNA staining of the infundibulum during the preovulatory phase. (**D**) SNA staining of the isthmus during the preovulatory phase. (**E**) High magnification of (**D**) showing the positivity of the cilia. (**F**) SNA staining of the isthmus during the luteal phase. Key: bv, blood vessel; ct, connective tissue; arrow, cilia. Scale bar: (**A**) = 12 µm; (**B**,**C**,**E**,**F**) = 10 µm; (**D**) = 50 µm.

**Figure 12 animals-12-02769-f012:**
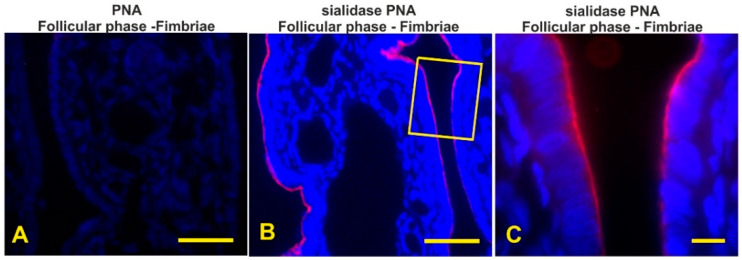
Localization of terminal and sialic acid penultimate Galβ1,3GalNAc by means of PNA (**A**) and sPNA (**B**–**F**) in the baboon *Papio hamadryas* oviductal epithelium during the menstrual cycle. (**A**) Incubation with PNA alone gave negative staining. (**B**) KOH-sialidase pretreatment revealed PNA binding sites on the luminal surface. (**C**) Detail of the squared zone in (**B**). (**D**) sPNA staining of fimbriae from preovulatory baboon. (**E**) Detail of sPNA staining in the preovulatory infundibulum. (**F**) sPNA staining of the isthmus from the luteal baboon. Scale bars: (**A**,**B**) = 50 µm; (**D**) = 100; (**C**,**E**,**F**) = 10 µm.

**Figure 13 animals-12-02769-f013:**
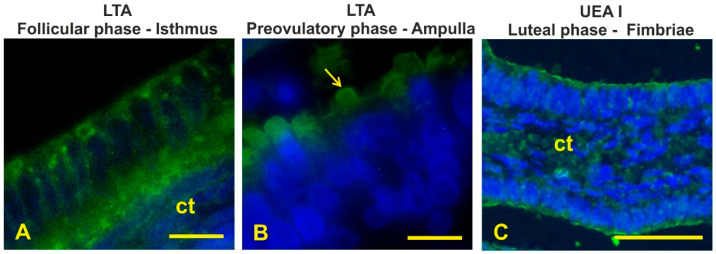
Staining of aL-Fuc terminating glycans with LTA (**A**,**B**) and UEAI (**C**) in the baboon *Papio hamadryas* oviductal epithelium. (**A**) LTA positivity of the isthmus during the follicular phase. (**B**) LTA positivity of the ampulla during the preovulatory phase. (**C**) UEA I binding sites in the fimbriae during the preovulatory phase. Note the presence of positive apical blebs in the ampulla from preovulatory baboons. Key: ct, connective tissue; arrow, apical protrusion. Scale bar: (**A**,**B**) = 10 µm; (**C**) = 50 µm.

**Figure 14 animals-12-02769-f014:**
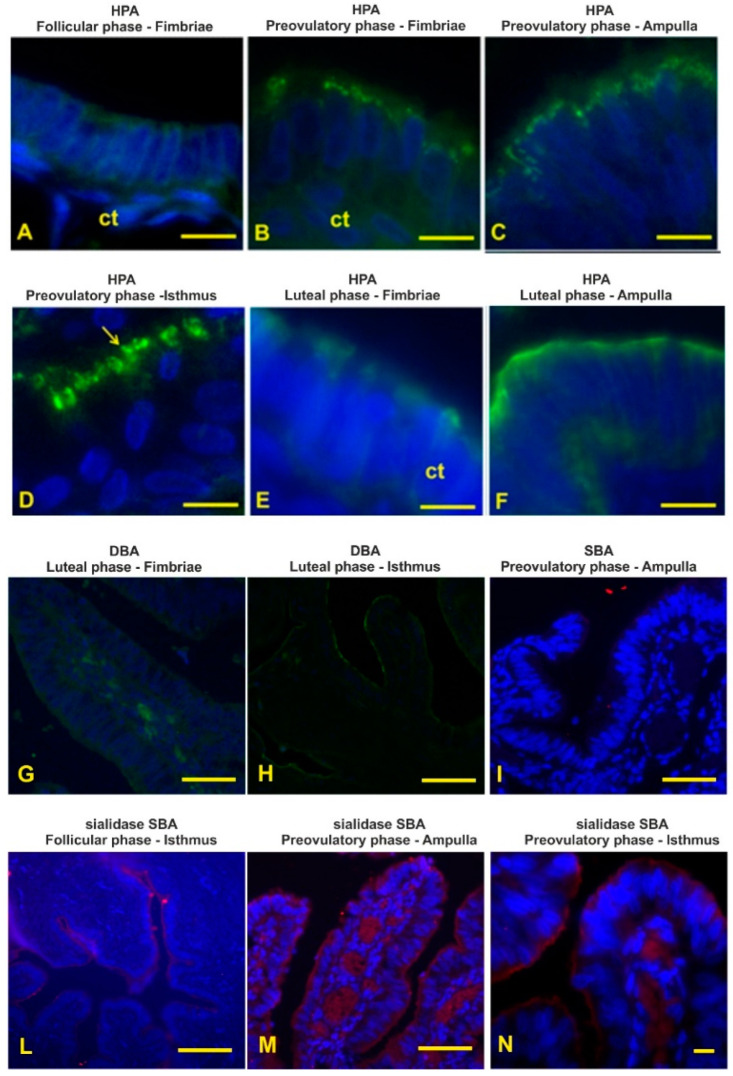
Localization of terminal GalNAc by means of HPA, DBA, SBA and sialic acid penultimate GalNAc terminating glycans by means of sSBA in the baboon *Papio hamadryas* oviductal epithelium during the menstrual cycle. (**A**) HPA-binding sites in the fimbriae during the follicular phase. (**B**,**C**) HPA reactivity with the fimbriae and the ampulla during the preovulatory phase. (**D**) HPA reactivity of the isthmus during the preovulatory phase. (**E**,**F**) HPA staining pattern of the fimbriae and ampulla during the luteal phase. (**G**,**H**) DBA reactivity of fimbriae and isthmus segments during the luteal phase. (**I**) SBA negative reaction with the oviductal epithelium (the picture shows fimbriae segment of a preovulatory animal). (**L**) Weak SBA staining of fimbriae from follicular baboon after KOH-sialidase treatment. (**M**,**N**) sSBA staining of the oviduct from preovulatory baboons. Key: ct, connective tissue; arrow, apical protrusion. Scale bar: (**A**–**F**) = 10 µm; (**G**–**I**,**M**) = 50 µm; (**N**) = 10 µm.

**Table 1 animals-12-02769-t001:** Lectin used, their sugar specificities, and the inhibitory sugars used in control experiments.

Lectin Abbreviation	Source of Lectin	Concentration (µg/mL)	Sugar Specificity	Inhibitory Sugar (0.5 M)
MAL II	*Maackia amurensis*	30	NeuNAcα2,3Galβ1-3GalNAc	NeuNAc
SNA	*Sambucus nigra*	30	Neu5Acα2,6Gal/GalNAc	NeuNAc
PNA *	*Arachis hypogaea*	30	Galβl,3GalNAc	Galactose
HPA	*Helix pomatia*	25	αGalNAc	GalNAc
SBA *	*Glycine max*	30	α/βGalNAc	GalNAc
DBA	*Dolichos biflorus*	30	αGalNAc	GalNAc
Con A	*Canavalia ensiformis*	25	αMan > αGlc	Mannose
LTA	*Lotus tetragonolobus*	30	αL-Fuc	Fucose
UEA I	*Ulex europaeus*	30	L-Fucαl,2Galβl,4GlcNAcβ	Fucose

Fuc, fucose; Gal, galactose; GalNAc, N-acetylgalactosamine; Glc, glucose; GlcNAc, N-acetylglucosamine; Man, mannose; NeuNAc, N-acetylneuraminic (sialic) acid. *, TRITC (rhodamine)-labeled lectins. Non-marked lectins were FITC (fluorescein isothiocyanate)-labeled lectins.

**Table 2 animals-12-02769-t002:** Lectin binding pattern of the epithelium lining the baboon *Papio hamadryas* oviduct during the menstrual cycle.

Lectin	Follicular Phase				Preovulatory Phase				Luteal Phase			
	Fim	Inf	Amp	Isth	Fim	Inf	Amp	Isth	Fim	Inf	Amp	Isth
Con A	c,ac,bl	c,as,bl	c,ac,bl	c,ac,bl	c,ac	ac	ac,ab	ac	c,ac,bl	c,ac,bl	c,ac,bl	c,ac,bl
MAL II												
sMAL II	-	-	-	-	-	-	-	-	-	-	-	-
SNA	-	-	±as	±as	sn,as	sn,as	sn,as	ci	ac,as	ac,as	ac,as	ac,as
sSNA	-	-	-	-	-	-	-	-	-	-	-	-
PNA	-	-	-	-	-	-	-	-	-	-	-	-
sPNA	as	as	as	as	as	as	as	as	as	as	as	as
HPA	±as,c	-	-	-	ac	ac	ac	ab	±as	as	as	as
DBA	-	-	-	-	±as	-	-	-	-	±as	-	±as
SBA	-	-	-	-	-	-	-	-	-	-	-	-
sSBA	±as	±as	±as	±as	as	as	as	as	-	-	-	-
LTA	-	-	-	± c,as	-	-	ab	as	-	-	-	-
UEA I	-	-	-	-	-	-	-	-	as	-	-	-

Key: ab, apical bleb; ac, apical cytoplasm; Amp, ampulla; as, apical surface; bl, basal lamina; c, entire cytoplasm; ci, cilia; Fim, fimbriae; Inf, infundibulum; Isth, isthmus; s, KOH-sialidase; sn, supranuclear cytoplasm. -, negative reaction; ±, faintly visible reaction. Unsigned results indicate clearly visible reaction.

## Data Availability

All data are available in the present manuscript.

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
