# Peer review of "Modification of Morphology and Glycan Pattern of the Oviductal Epithelium of Baboon Papio hamadryas during the Menstrual Cycle"

_animals, 2022, doi:10.3390/ani12202769_

Round 1

Reviewer 1 Report (Previous Reviewer 3)

The authors addressed most of the points suggested in my previous review in the revised version of the manuscript.

Author Response

Dear reviewer, Thank you very much for taking your time to review our revised manuscript.

Reviewer 2 Report (Previous Reviewer 2)

The authors made all the changes suggested in the first revision of this manuscript. The new version of the manuscript is suitable for publication.

Author Response

Dear reviewer, Thank you very much for taking your time to review our revised manuscript.

Reviewer 3 Report (Previous Reviewer 1)

This manuscript has improved. There are two contradictions of methods with the results/objective of the study that need clarification. One is the analysis and presentation of the isthmus region. The other is inclusion of impubertal animals in the study.

There are places in the manuscript where I suggest reorganization of sentences to aid the readers or flag a typo.

Author Response

Dear Reviewer, Thank you very much for taking your time to review our manuscript (animals-1941473). We really appreciate all your comments and suggestions. These comments and suggestions have enabled us to improve the quality of our manuscript. We have revised the manuscript according to your suggestions and highlighted in red all changes in the revised manuscript. Point by point responses comments are listed below. We hope the revised manuscript has addressed all concerns. Kind regards, Salvatore Desantis Corresponding author Reviewer - Comments and Suggestions for Authors This manuscript has improved. There are two contradictions of methods with the results/objective of the study that need clarification. One is the analysis and presentation of the isthmus region. The other is inclusion of impubertal animals in the study. Response: • This research investigated the oviduct from the fimbriae to isthmus. Since the utero-tubal junction constitutes the oviduct, in Materials and Methods section (2.5) we specified that this segment was not analyzed because it was damaged during the salpingectomy. • In this research were used sexually mature female and the reference to impuberal subjects was a pure mistake. Thus, the reference to impuberal subject was deleted in the revised text. There are places in the manuscript where I suggest reorganization of sentences to aid the readers or flag a typo. Lines 115: “disperses” replaced with “dispersed”. Line 116 (old version): “although is a prey item” was deleted. Lines 136-147 (old version): the text was modified as suggested (lines136-142, new version). Line 158 (old version): “impuberal subject” was added by mistake and “adult” was replaced with “sexually mature” (line 155, new version). Line 164 (old version): “mature sexual subjects” was replaced as suggested “sexually mature subjects” (line 161, new version). Lines 200-202 (old version): in this research the utero-tubal junction was not used because damaged during the salpingectomy. This was better specified in the revised version (line 198, new version). Line 243 (old version): “CaCl2” was corrected “CaCl2” (line 239, new version). Line 538 (old version): “lower” replaced with “lowest” (line 532, new version). Line 547 (old version): “a pressure that produces” was removed as suggested. Line 549 (old version): the article was added as suggested (line 555, new version). Lines 551-552 (old version): the sentence was incorporated into the successive paragraph as suggested (line 544-545, new version). Line 563 (old version): as suggested we specified that sSBA staining refers to “fimbriae and all other segments of the oviduct” (line 555, new version).

This manuscript is a resubmission of an earlier submission. The following is a list of the peer review reports and author responses from that submission.

Round 1

Reviewer 1 Report

This article provides the histological evaluation of the oviduct of baboons. Specifically evaluated were the epithelial cell heights of four anatomical regions through the three identified stages of the follicular cycle. The authors also investigated the epithelial expression and localization of glycans using lectin affinity assays. Finally, the authors evaluated the stage in the follicular cycle with a combination of vaginal cytology and hormone assays.

The introduction of the subject supports the evaluation of the morphology of the baboon oviduct but does not stress the value added to the literature by the addition of this information of a separate species of baboon. A few sentences on mating patterns or other differences between the species would aid the reader. The reader may also be distracted from the work by the use of language that indicates tiers to the value of primates (subhuman rather than non-human.)

Methods:

The presented methods are clear and well written. However, there are gaps in the information regarding the following:

1)      Describe how the oviducts were oriented within the block for sectioning. Cross sections or longitudinal?

2)      Describe how the anatomical regions were identified on the slides.

3)      Describe the replication scheme for lectin staining.

4)      Describe the number of fields read for each sample and how they were selected.

5)      Describe the judgement of +, weak or negative staining was evaluated.

a.       How many readers were there?

b.       How many sections were evaluated per animal?

c.       How was staining intensity decided when animals differed in intensity within a region and within a follicular stage?

Results:

Figures 5,8-11. Include information regarding what region of the oviduct each image represents on the borders. (X = region, Y = stage or vice versa). Figures 5, 8-11 change the titles to reflect what glycan structures are represented.

Throughout the results with each lectin section. Introduce the glycans that are being discussed. Include something about what these types of glycans are participating in within the oviduct.

Discussion:

Please condense the discussion section to aid the reader.

For example, the paragraph linking vaginal smear morphology to hormone levels is information that is well established and does not need extensive discussion.

The authors seem to be focused on the lectins ability to bind to sections, but the interesting portion of this study is the glycan structures that are present. Revise throughout the manuscript to refocus on the glycans rather than the lectins. Please apply information from the review by A Varki “Biological Role of Glycans” published 2017 in Glycobiology.

Reviewer 2 Report

The manuscript “Modification of morphology and glycan pattern of the oviductal epithelium of baboon Papio hamadryas during the menstrual cycle” aimed to examine the morphology and the glycan composition of the oviductal epithelium of baboon Papio hamadryas during the menstrual cycle. The manuscript is interesting and some comments/suggestions, I enumerate below:

In the simple summary and abstract:

1. In the phase: “These results provide an insight into molecular differences of the baboon oviduct...”, do the authors need to detail the relation to which they are talking about molecular differences, according to the phase of the cycle? Of region? The information is incomplete as it is.

2. Review the formatting of the abstract.

In introduction:

1. The introduction was well written, and the authors clearly addressed the problem of the study. I only suggest that the authors explore further in talking about the species of study, commenting on its world population, and its economic, ecological, and scientific importance.

In the material and methods:

1. What would be a “sub adult subject”? Detail in the manuscript.

2. “without endometrial disorders (hyperplasia, endometritis, neoplasia)”, detail how this assessment was performed for the study.

3. Inform at what time the collections for hormonal analysis were performed. Were they performed more than once? At what intervals?

4. “For the measurement we selected…”, correct.

5. I suggest separating the morphometric analysis from the statistical analysis and that the morphometric analysis has more details, with for example: magnitude used, number of fields employed, etc.

In the results and discussion:

1. Detail on “A region-specific glycosylation pattern was also detected.”

2. Figure scales must be corrected from ","to "."

Reviewer 3 Report

In the present manuscript the authors analyze the histological changes produced in the oviductal epithelium during the menstrual cycle of the baboon Papio hamadryas. Additionally, the lecting binding pattern was also described. The information provided in this study is original because this species was not analyzed previously. The study provides new and relevant information about the changes produced in the epithelium and the glycoproteins present in the different regions of the oviduct showing differences among species and can be related with different relevant physiological process that happen in the oviduct as the fertilization, sperm reservoir and early embryo development.

I consider that this article can be published after the appropriate changes that I will described below. 

MAJOR COMMENTS

.- It is necessary to improve the references used in the introduction section.

.- The authors make reference about the specific oviductal glycoproteins produced in the ovidcut; however, they do not use a specific antibody against this glycoprotein. There are different antibodies commercially available. 

.- Other lectins specific for N linked oligosaccharides is not used. No lectins specific for Galbeta1,4GlcNAc as DSA or RCA I were used in this study. These sequences are typical of complex N linked oligosaccharides. Fucose residues linked to the core mannose of N-linked can be detected using AAA lectin. 

.- The authors should focus on their finding and not on the oviductin, aquaporin o extracellular vesicles because they do not analyze them with specific tools.

MINOR COMMENTS

Introduction section

1.- Lines 74, 88 and 91. More recent references should be included.

2.- Line 95-96. Please add references.

3.- Line 99. More references are required. It is important to take in consideration that the mice KO of oviductin is fertile, a seudogenization process is produced in other species. Please revise this reference doi: 10.1007/s00239-018-9878-0 from Monget lab.

4.- Line 100. Please provide some references supporting the sentence.

5.- Line 112. In this study the authors do not analyze the glycan composition of the oviductin. They analyze glycoproteins present in the epithelium. For example the lectin binding detected in ciliated cells will not correspond to the oviductin. Please modify this sentence.

6.- Line 113. References should be provided.

7.- Lines 114-116. It is important to clearly indicate if the information is related with the oviductin or glycoproteins in general. The use of OGP is confusing. 

Material and Methods section

8.- Line 151. I consider that the reference number 17 is not appropriate because it does not correspond with the baboon. Please use specific references for the baboon species. 

9.- Line 153. Is there any information more precise of the day of cycles of the samples obtained?

10.- Line 166. I consider that it is better to change the black line that marks the ampulla. Probably you can use a dashed line at the isthmus and infundibulum region and indicate that the ampulla region of the oviduct is limited by the dashed line.

11.- Line 176 and Table 1.  The concentration of the inhibitory sugar used for each lectin should be included in the table. A image of the negative results obtained should be included. 

The lectins can have affinity for some sugars like Glc in the case of ConA but it is known that these sugars are not presented in the glycoproteins. 

Please revise the name of the SNA lectin.

12.- A neuraminidase treatment for the sialic acid lectins is strongly recommended.

Results section

13.- Figure 2. I consider that the color bars should not be superimposed. The error bar in the case of estradiol in the luteal phase cannot be visualized.

14.- Line 229. The oviduct is formed by three layers. Only two layers are mentioned.

15.- Line 231. In relation with the diameter of the ampulla, it is not clear if the authors are talking about the lumen of the ampulla or the entire oviduct conduct in this region. 

16.- Lines 232-233. In this case and in the previous comment it is recommended to make a quantitative analysis of the thickness as performed for the epithelial thickness.

17.- Lines 239-240. This sentence is contradictory. It is not possible to say that an epithelium is columnar and the cells are cuboidal. 

No information is provided about the percentage of ciliated cells in the different region of the oviduct in the preovulatory phase.

I strongly recommend the authors to use a specific antibody against cilia. 

18.- Figures 6 and 7. Please revise the spelling of the fimbriae and Isthmus.

19.- Lines 282-283. I cannot observe a positive basal lamina in the figure 8C. The basal lamina is not clear in the figure 8A. 

20.- Line 297. Please check the spelling of the basal lamina

21.- Table 2. It is necessary to use positive control for the lectins PNA and SBA. I recommend the use of neuraminidase treatment before the use of these lectins. Please revise the SBA in the luteal phase, in the case of the Isthmus, it is empty. 

22.- Line 323 and Fig 10D. It is not clear that the staining is associated with the apical blebs. 

23.-Lines 320-322. No clear differences are observed between the figure 10B and 10C. 

24.- figure 10G. A faint staining seem to be present in the oviduct. Why the fluorescence is now red?

25.- Line 341-343. I recommend the use of the pre-treatment of the sections with the enzyme neuraminidase. The T-antigen is usally masked by the neuraminic acid. This result can be compared to the MAL II lectin.

26.- Figure 9E. In the image it is not possible to recognize the cilia. 

27. Line 345. The staining of the figure 11A do not correspond to the text.  The staining is much stronger than mentioned.

28.- Lines 353-356, figure 11. A description of the figures is required.

Discussion section

29.- Lines 364-365. Is there any differences among baboon species of the age of puberty and menstrual cycle length?

30.- Lines 381-382. These sentences are more appropiated to mention before, for example at level of line 370.

31.- Line 412. The oviductal secretory cells are merocrine cells. They are not apocrine cells.

32.- Lines 413-415. Please provide references.

33.- Lines 440-443. This staining could also correspond to the Golgi apparatus. 

34.- Lines 453-455. This sentence is not true. You can find figures and tables of other regions of the oviduct in this article. 

35.- Lines 456-468. The authors focus on OGP or oviductin and they do not use specific tools to detect this mucin. Similar comment for other parts of the discussion. The authors should focus on their own data and compare to previous similar studies.

36.- Lines 495-497. It is no clear why the authors include these sentences. 

37.- Liness 501, 505 and 558. I do not agree about the apocrine secretion. Previously the authors talk about merocrine secretion (Line 478). 

38.- Lines 544-545. This process does not happen in the isthmus.